# Food Habits: Insights from Food Diaries via Computational Recurrence Measures

**DOI:** 10.3390/s22072753

**Published:** 2022-04-02

**Authors:** Amruta Pai, Ashutosh Sabharwal

**Affiliations:** Scalable Health Labs, Department of Electrical and Computer Engineering, Rice University, Houston, TX 77005, USA; ashu@rice.edu

**Keywords:** habitual behavior, food diaries, food habits, food consumption, MyFitnessPal, recurrent foods, food choices

## Abstract

Humans are creatures of habit, and hence one would expect habitual components in our diet. However, there is scant research characterizing habitual behavior in food consumption quantitatively. Longitudinal food diaries contributed by app users are a promising resource to study habitual behavior in food selection. We developed computational measures that leverage recurrence in food choices to describe the habitual component. The relative frequency and span of individual food choices are computed and used to identify recurrent choices. We proposed metrics to quantify the recurrence at both food-item and meal levels. We obtained the following insights by employing our measures on a public dataset of food diaries from MyFitnessPal users. Food-item recurrence is higher than meal recurrence. While food-item recurrence increases with the average number of food-items chosen per meal, meal recurrence decreases. Recurrence is the strongest at breakfast, weakest at dinner, and higher on weekdays than on weekends. Individuals with relatively high recurrence on weekdays also have relatively high recurrence on weekends. Our quantitatively observed trends are intuitive and aligned with common notions surrounding habitual food consumption. As a potential impact of the research, profiling habitual behaviors using the proposed recurrent consumption measures may reveal unique opportunities for accessible and sustainable dietary interventions.

## 1. Introduction

Human behavior exhibits both exploration and exploitation tendencies [1]. Consumption, broadly defined across many behaviors, is often characterized as a combination of novel (exploration) and repetitive events (exploitation) [2]. Similarly, in the domain of food consumption, we can characterize human behavior associated with food choice selection as a mixture of novelty-seeking choices and habitual choices [3,4,5]. The novelty-seeking consumption corresponds to new or occasional food items consumed by an individual. The exploratory behavior of the individual drives novelty-seeking consumption. Habitual consumption signifies food choices that are repeated regularly, often under similar contexts [4]. The exploitation behavior of the individual drives the habitual consumption. Both parts of human behavior provide insights into the individuals’ food selection behavior.

Habitual consumption has traditionally been measured with food frequency questionnaires (FFQ) [6], or 24-diet recalls. FFQ suffers from high recall bias [7], and as food lists are culture-specific, predefined food lists need to be modified and validated for use in different contexts. Alternatively, 24-h diet recalls, and 7-day food diaries provide more detailed information at the cost of administration time and the burden of trained interviewers [8]. Recently, smartphone-assisted food records have been discussed as an innovative technology for diet measurement in epidemiological studies [9]. Diet self-monitoring smartphone applications have grown popular over the last decade with increasing attention towards a quantified self. Some of the app-based services have demonstrated clinical impacts [10,11,12].

Smartphone-based applications [13] for daily food logging are also emerging as a promising instrument for understanding food consumption behaviors [14]. In recent works [5,15], the authors utilized food diaries to study and model food consumption behavior. Related important past works [5] studied the repeat consumption of food choices and created prediction models to forecast the top-k food choices each day for users of a food diary dataset [16]. Another insightful study [15] used the daily logged data of individuals to recommend easy to adopt personalized food suggestions for behavior change by leveraging frequent past healthy behaviors and occasionally exploring infrequent behaviors, while past works [5,15] have demonstrated the utility of habitual food consumption behavior. We identified four research gaps in the current understanding of habitual food consumption.

*Characteristics of habitual consumption:* Previous work [5] discussed repeat consumption in the context of habitual consumption and defined repeat consumption behavior as the average fraction of items consumed in a day that was consumed in the last seven days. Since repeat consumption captures short-term repetitions, it is a short-term characterization of habitual consumption. Certain food choices will occur more than once in the food consumption sequence.

However, not all food choices that occur more than once are habitual. We developed measures that define habitual consumption based on both short- and long-term occurrences in the food consumption sequence. However, food consumption sequences collected through food diaries present only a snapshot of an individual’s overall food consumption behavior. Hence, the estimation of the occurrence characteristics depends on the length of the consumption sequence available for computation. The reliability of the habitual consumption measure also depends on the length of the consumption sequence.

*Food-item and meal habitual consumption:* Food choices can be described as individual food-items or as a meal, which is a combination of individual food-items. An example of habitual consumption behavior for individual food-items is snacks, often consumed in-between meals. In contrast, an example of habitual meal consumption is packaged meals in fast-food restaurants, e.g., the main food-item, sides, and a drink. Hence, a study of habitual consumption should distinguish between food-items and meals, as meals comprise of food-items. The number of food-items combined in a meal varies for each individual and over time. The number of food-items chosen in a meal may influence habitual food selection behavior.

*Context of habitual consumption:* Nutrient consumption differs for meal occasions [17]. Individuals eat different types of food-items at each meal occasion, and thus habitual behavior in food selection may differ for meal occasions. Another context is weekend versus weekend dietary behaviors. Individuals are known to follow more routine behaviors during the week and exhibit more exploratory behaviors during the weekend.

*Association with food logging duration:* Food logging is in itself an intervention. A common feature present in the food logging app is that it allows individuals to log historical food choices more quickly. Thus, individuals who use the app for a longer duration may repeat or log habitual food choices because it is easier to record in the app.

**Contributions**: To address the above research gaps, we aim to characterize habitual consumption behavior from food diaries systematically. Towards that end, we make the following four contributions:

*Computational measures of habitual consumption*: We introduced recurrence as a principal characteristic of habitual consumption. Consequently, we define *recurrent consumption as a new behavioral measure for habitual food consumption*. We computed recurrent consumption measures from longitudinal food diaries. The proposed measures are:*Recurrent choices*: set of regularly repeated food choices. This provides information on which items are habitual and may be targeted to modify eating habits.*Recurrence strength*: fraction of consumption of the recurrent choices. This signifies the strength of habitual consumption exhibited by an individual. This indicates the habit-forming capability of the individual and can be used to profile the habitual consumption behavior of individuals.

We analyzed trends in recurrence consumption of MyFitnessPal users of a public food diary dataset [18].

*Recurrent consumption of food-items and meals*: We investigated the recurrent consumption of individual food-items and meals. Towards that end, we introduced the recurrence strength tuple, namely food-item recurrence strength, food-item-per-meal recurrence strength, and meal recurrence strength, and these capture the degrees of habitual food consumption behavior. We analyzed the correlations between the recurrence strength tuple and the average number of food-items consumed per meal. We found that recurrent consumption of food-items was higher than the recurrent consumption of meals. Individuals that consume many food-items in a meal exhibit lower recurrent meal consumption than individuals who choose from a small number of food-items.

*Contextual recurrent consumption*: We used the recurrence strength tuple to analyze the habitual consumption behavior of users from the MyFitnessPal food diary dataset for various contexts. We found that the recurrence strength was highest for breakfast and lowest for dinner. We observed that weekday recurrence strength is higher than weekend recurrence strength. The recurrence strength during weekdays and weekends was highly correlated, suggesting that individuals with relatively high recurrent consumption on weekdays also have relatively high recurrent consumption on weekends.

*Association between recurrent consumption and food logging*: To investigate if food logging duration is associated with recurrent consumption, we presented the correlation analysis between recurrence strength and the number of recorded days. We found an insignificant correlation between the number of recorded days and recurrence strength, implying that food logging does not influence recurrent consumption behavior.

One of the primary barriers to healthy eating is taste and difficulty in changing eating habits [19,20]. Additionally, recent research in food recommendation, such as the Cultural Double Pyramid [21] suggests that sustainable diets can reflect local tradition, culture, and preferences. Understanding an individual’s habitual choices and potentially designing interventions around those habitual choices could be a long-term sustainable strategy. Our methodology can also be used alongside current diet adherence score models [22] to evaluate an individual’s diet quantitatively and identify personalized opportunities for effective food interventions that result in long-term habit change.

## 2. Material and Methods

The objective was to quantify an individual’s habitual food consumption behavior. We examined the food selections individuals make in their daily lives to estimate their habitual choices. Thus, the signal of interest is a sequence of meals consumed by the individual in a given observation duration with a label describing the meals’ occasion and text describing the names of the food-items consumed during the meal. We extracted the signal from food diary logs. We used a public food diary dataset [5,18] of MyFitnessPal users to investigate the signal.

### 2.1. Myfitnesspal Food Diary Dataset

We used the MyFitnessPal [13] food diary dataset created by Weber and Achananuparp [18]. The dataset contains 587,187 days of food diary entries collected from 9896 individuals spanning six months from September 2014 to April 2015. The participating individuals belong to an online weight loss community and used MyFitnessPal to log their food. A typical food diary entry of a day consists of textual and nutritional descriptions of the meals consumed. The descriptions contain the following information: (i) de-identified participant number, (ii) date of a food log, (iii) the meal occasion label input by user or default set by the MyFitnessPal application(e.g., breakfast, lunch), (iv) the names of food-items consumed in the meal, (v) nutrition of each food-item (e.g., calories, protein, fat), and (vi) MyFitnessPal app provided calorie goal for the day. Figure 1 shows the current version of the app interface [13].

An example of one whole day of food record data for a user is provided in the Appendix A. More details on the creation of the dataset by the authors is provided in their work [18]. Additionally, the authors released demographic information on gender, age group, and region in the United States of the users in the MyFitnessPal dataset in their subsequent work [5]. The available demographic statistics of the users in the dataset are presented in Table 1. The dataset is highly skewed towards females and the 18–44 years age group.

### 2.2. Preprocessing of MyFitnessPal Food Diary Dataset

The food diary records have extra information, e.g., serving size, nutritional information, along with our signal of interest. Therefore, we preprocessed the data to remove irrelevant information to our analysis. The food diary logs are irregularly spaced because the recording requires a high individual engagement that is burdensome to sustain over long periods [23]. The mean number of days where at least a single meal occasion is recorded is 59 days (S.D = 55, median = 42, max = 187 , min = 1, N = 9896). Hence, we preprocessed the data and include a subset of individuals with complete and consistent records to avoid the influence of missing information in our analysis. In the following subsections, we describe the preprocessing steps.

#### 2.2.1. Removing Serving Size and Nutrition Information

We excluded the serving size data, energy, and nutrition content information because our focus was on studying habitual behavior in food selection, as captured by names of the food-items.

In the MyFitnessPal dataset, the serving size description is part of the text describing the food-item. Fortunately, the data format separates the food-item name and serving size descriptions with a comma. Hence, we excluded the serving size description using the string split method. While we did not use the energy content information in habitual behavior analysis, we used it to preprocess the dataset as described below.

#### 2.2.2. Removing Non-Descriptive Items

We removed meals with entries that are described as “quick added calories” as this does not contain information about the food-item consumed. We removed meals where the calorie value was greater than 3000 kcal, assuming such entries resulted from errors while logging [5]. In the included meals, we removed nonfood entries, such as water, medications, and supplements by removing entries that have a calorie amount of zero.

#### 2.2.3. Removing Custom Meal Occasion Labels

The MyFitnessPal app provides users with the option to use default labels or create custom labels for meal occasions. The default labels are *breakfast, lunch, dinner, and snacks*. As our analysis consists of comparisons across meal occasions, we retained individuals that used the default labels and excluded individuals with custom labels.

#### 2.2.4. Removing Days with Missing Meals

One of the challenges of self reported data is the prevalence of missing information in the dataset due to irregular logging. We took the following steps to reduce the impact of missing information in our analysis. We included users who were complete and consistent in their food records. For each individual, complete days that are days with an entry for all meal occasions were included and all other days excluded. Individual that had recorded complete days consistently were included, i.e., the average number of missing days between the complete days ≤3 (less than half a week).

We did not use data imputation to fill the missing information because our habitual behavior analysis is reliant on the repetition properties of food choices. Imputing missing meals with frequently occurring food-items or food-items proximal in the sequence would artificially inflate the recurrent strength computation and bias the results. Additionally, imputation methods for complex categorical sequence of sets that involves predicting the number of food-items in the meal (size of the meal set), followed by the group of food-items that would occur together in the meal (elements of the set) are not well established.

After performing the preprocessing steps described above, we retained data of 2758 individuals. For each individual, we have the food diary information for a certain number of days. The number of days ranges from 1 day to 180 days across individuals. An individual’s food diary record for a particular day consists of four entries (one for each meal occasion). Each entry presents a list of food-items that were consumed at the meal occasion. As we did not have information about the time of consumption for each food-item, we assumed the food-items under a meal occasion were consumed in combination as a meal.

Thus, for each individual, there are four food consumption sequences, one for each meal occasion. Since the days considered for each meal occasion is the same within an individual, the length of the food consumption sequence for each meal occasion is equal for an individual. Each consumption sequence is a sequence of meals consumed by the individual. Each meal is a combination of food-items.

### 2.3. Computation of Recurrent Consumption

In this section, we introduce definitions for recurrent consumption. We describe the computation of recurrent consumption measures, i.e., recurrent food-items, recurrent meals, and recurrence strength from the food consumption data of the individual.

#### 2.3.1. Notation

Let *u* denote the individual. Each individual logs their meal and denotes the meal occasion. We denote the meal occasion as *o* and focus our attention on four meal occasions, o∈{breakfast,lunch,dinner,snacks}. Each meal occasion can consist of multiple food-items. Let *d* denote a food-item. Let *D* be the set of distinct food-items for all the participants in the dataset. Thus, *D* is the food-item library comprising all food-items logged in the dataset and thus d∈D. Let $D_{u,o}$ denote the set of distinct food-items consumed by individual *u* during meal occasion *o*. We refer to $D_{u,o}$ as the *food-item library of the individual**u* for meal occasion *o*.

A meal *m* is a set of one or more food-items, and thus m⊂D. Let *M* be the set of distinct meals in the dataset; therefore, *M* is a set of sets. Let $M_{u,o}$ be the set of unique meals consumed by an individual *u* during meal occasion *o*. We refer to $M_{u,o}$ as the *meal library of the individual u* for meal occasion *o*. Let $m_{u,o,t}\in M_{u,o}$ denote the meal consumed by an individual *u*, at meal occasion *o* on day *t*; note $m_{u,o,t}\subset D_{u,o}$.

For an individual during a meal occasion, the extracted signal is a food consumption sequence of meals written as ${\left[m_{u,o,t}\right]}_{t=1}^{t=n_{u}}=\left[m_{u,o,1},m_{u,o,2},m_{u,o,3},......,m_{u,o,n_{u}-1},m_{u,o,n_{u}}\right]$, where *u* refers to the individual, *o* refers to the meal occasion, *t* refers to the the $t^{\mathrm{th}}$ day of the sequence, and $n_{u}$ refers to the length of the sequence of individual *u*.

Note that the sequence of days may not be temporally consecutive, as there may be missing days in a user’s log. Additionally, note that each user may have a different number of days $n_{u}$ in the sequence. The number of days could have been a function of the meal occasion as well. However, since in preprocessing, we included only those days with entries for all meal occasions, $n_{u}$ is only individually varying.

Consider the simplistic example shown in Table 2. We created 14 days of breakfast consumption sequence for an individual with u=1. The days are not temporally consecutive. From the meal sequence presented in Table 2, we identify that there are four possible food-items: latte, croissant, muffin, and hot-chocolate. The four food-items form the food-item library $D_{u,o}$. There are four possible meals or food-item combinations indicated by the meal library $M_{u},o$. The meals are latte and muffin, latte and croissant, latte by itself, or muffin and hot chocolate.

#### 2.3.2. Definitions

In this section, we introduce the key definitions to quantitatively capture recurrent consumption. Food-item $d\in D_{u,o}$ or meal $m\in M_{u,o}$ consumed by the individual *u* during the meal occasion *o* occurs in the sequence ${\left[m_{u,o,t}\right]}_{t=1}^{t=n_{u}}$. Let $T_{u,o,d}$ denote the set of days that food-item *d* occurred in the sequence ${\left[m_{u,o,t}\right]}_{t=1}^{t=n_{u}}$. Let $T_{u,o,m}$ denote the set of days that meal *m* occurred in the sequence ${\left[m_{u,o,t}\right]}_{t=1}^{t=n_{u}}$. The set of days of occurrences of food-item and meal are computed as $T_{u,o,d}=\left\{t|d\in m_{u,o,t}\right\}$ and $T_{u,o,m}=\left\{t|m_{u,o,t}=m\right\}$, respectively. There is a need to study the occurrences characteristics of both food-items and meals to understand habitual consumption. We introduce two principle characteristics, the relative frequency and span that are used to differentiate between habitual and non habitual food-items or meals.

**Definition** **1**(Relative frequency)**.**
*We define relative frequency of a food-item as the normalized number of occurrences of the food-item in the sequence. Let $f_{u,o}\left(d\right)$ denote the frequency (number of occurrences) of food-item d in ${\left[m_{u,o,t}\right]}_{t=1}^{t=n_{u}}$. The relative frequency of d in ${\left[m_{u,o,t}\right]}_{t=1}^{t=n_{u}}$ is denoted as $\frac{f_{u,o}\left(d\right)}{n_{u}}$; note $f_{u,o}\left(d\right)=\left|T_{u,o,d}\right|$. Similarly, the relative frequency of a meal is written as $\frac{f_{u,o}\left(m\right)}{n_{u}}$, which denotes the number of occurrences of meal m in ${\left[m_{u,o,t}\right]}_{t=1}^{t=n_{u}}$ with $f_{u,o}\left(m\right)=\left|T_{u,o,m}\right|$ .*

Intuitively we expect that habitual food-items or meals will have a high number of occurrences in an observed duration of time. Individuals used the MyFitnessPal app for varying durations. Hence, the length of the observation duration is not the same among individuals. Consequently, the length of the sequence ${\left[m_{u,o,t}\right]}_{t=1}^{t=n_{u}}$ will not be the same. For individuals with the same observation duration, the sampling of the observation duration may not be uniform. The observation duration is regularly sampled, i.e., each day is recorded for certain individuals. For others, the duration is irregularly sampled, i.e., missing days between recorded days.

Hence, individuals with the same observation duration may have a different number of days in the sequence. Thus, studying the number of occurrences alone is not feasible. Hence, we performed normalization by taking the ratio of the number of occurrences and sequence length $n_{u}$. Interestingly, the ratio can be interpreted as a consumption rate, i.e., how often an item is consumed per day. While discussing habits, we often describe it as a rate, for example, *“I shower daily or I go to the gym every day”*. Hence, quantifying the consumption rate of a food-item or meal provides a meaningful characterization. For example, for an individual who eats cereal daily, the relative frequency of cereal would be 1 (once per day).

**Definition** **2**(Span)**.**
*We define the span of the food-item in a given food consumption sequence as the normalized difference in days between the first and last occurrence of the food-item/meal in the sequence ${\left[m_{u,o,t}\right]}_{t=1}^{t=n_{u}}$. The span of food-item d is given as $\frac{\left(\max\left(T_{u,o,d}\right)-\min\left(T_{u,o,d}\right)\right)}{n_{u}}$. Similarly, the span of a meal m is given as $\frac{\left(\max\left(T_{u,o,m}\right)-\min\left(T_{u,o,m}\right)\right)}{n_{u}}$.*

The span measures the time difference between the food-item’s or meal’s first occurrence and the last occurrence in the sequence. A larger number means that the food choice occurred across a longer period, which signifies that the individual selected it across a longer duration. Much like other human consumption behaviors, boredom can affect food choice selection. Hence, we expect that food-items may be frequently consumed until boredom occurs, and then the consumption is discontinued. However, intuitively habitual food-items are consumed despite boredom or are frequently re-consumed after a period of discontinuity.

Either way, habitual food-items should span a major portion of the observation duration. Hence, *span* quantifies the spread of the occurrence of a food-item or meal in the observation duration. For example, in an observation duration of one month (28 days) with each day recorded, a food-item was consumed twice in the first week on Monday and Tuesday and consumed thrice in the last week on Monday, Tuesday, and Thursday. In the example, the span of the food-item would be $\frac{16}{28}$. In a contrasting example, the considered food-item was consumed five times in the first week alone from Monday to Friday and not consumed again for the rest of the month. In the second example, the span would be $\frac{4}{28}$. Note that, by definition, the relative frequency of the food-item in both examples would be $\frac{5}{28}$.

Intuitively, we would consider a food selection as recurrent if the choice is repetitive (“high” frequency) and appears across a major portion of the observation duration (“large” span). We use this intuition to define recurrent consumption. We impose criteria on both the frequency and span characteristics of the food-item or meal. The frequency criterion is to select items repeated a significant number of times in the observed duration. In our analysis, we set the threshold for the frequency criterion based on the observation that people generally shop groceries weekly, and familiarity is a crucial driver during food selection [24].

Thus, the minimum relative frequency of recurrent items is one per week, i.e., $\frac{1}{7}$ per day. The span criterion is to select items that have occurrences spread out in the sequence. Items that are highly repeated in a particular week but are not repeated in other weeks may satisfy the frequency criterion but are not habitual food-items. To avoid selecting such locally repeated food-items as recurrent food-items, we impose the span criterion of $\frac{1}{2}$. The span criterion is that the first the last occurrence should be at a minimum half the sequence length apart.

Using the criteria as mentioned earlier, we define the food-items and meals in the individual’s food-item library $D_{u,o}$ and meal library $M_{u,o}$, respectively, that are recurrent based on the relative frequency and span of their occurrence in the food sequence. A recurrent choice (food-item or meal) in a sequence is a food choice that frequently occurred across a large span.

**Definition** **3**(Recurrent choice)**.**
*A recurrent choice in a food consumption sequence is a choice (food-item or meal) that has a relative frequency of greater than $\frac{1}{7}$ and a span of greater than $\frac{1}{2}$ in the sequence.*

We compute two sets, namely the recurrence food-item set ${\tilde{D}}_{u,o}$ and recurrent meal set ${\tilde{M}}_{u,o}$ that consist of recurrent food-items and recurrent meals, respectively. The computation of the recurrent food-item set and recurrent meal set is described below.

*Recurrent food-item set, ${\tilde{D}}_{u,o}$*: Recurrent food-item set ${\tilde{D}}_{u,o}$ is the set of all recurrent food-items occurring in the food consumption sequence ${\left[m_{u,o,t}\right]}_{t=1}^{t=n_{u}}$ of individual *u* for meal occasion *o*. The recurrent food-item set is
(1)$$\begin{array}{c} {\tilde{D}}_{u,o}=\left\{d\;|\;\frac{f_{u,o}\left(d\right)}{n_{u}}\geq \frac{1}{7},\frac{\left(\max\left(t_{u,o,d}\right)-\min\left(t_{u,o,d}\right)\right)}{n_{u}}\geq \frac{1}{2}\right\}. \end{array}$$*Recurrent meal set, ${\tilde{M}}_{u,o}$*: Recurrent meal set ${\tilde{M}}_{u,o}$ is the set of all recurrent meals occurring in the food consumption sequence ${\left[m_{u,o,t}\right]}_{t=1}^{t=n_{u}}$ of individual *u* for meal occasion *o*. The recurrent meal set is
(2)$$\begin{array}{c} {\tilde{M}}_{u,o}=\left\{m\;|\;\frac{f_{u,o}\left(m\right)}{n_{u}}\geq \frac{1}{7},\frac{\left(\max\left(t_{u,o,m}\right)-\min\left(t_{u,o,m}\right)\right)}{n_{u}}\geq \frac{1}{2}\right\}. \end{array}$$

We introduce recurrent consumption as a measure of habitual consumption. Recurrent consumption refers to consumption of recurrent items. Recurrence strength is the proportion of recurrent consumption in an individual’s total consumption. For an individual *u* and meal occasion *o*, we define *food-item recurrence strength*, *food-item-per-meal recurrence strength* and *meal recurrence strength* denoted by three as $\alpha_{u,o},\beta_{u,o},\gamma_{u,o}$. We refer to the three recurrence strength measures, $\left(\alpha_{u,o},\beta_{u,o},\gamma_{u,o}\right)$ as the recurrent tuple for individual *u* at meal occasion *o*. The recurrence strength tuple summarizes the levels of habitual food consumption exhibited for individual food-items and meals.

**Definition** **4**(Recurrence strength tuple)**.**
*We define three quantities to capture the* recurrence strength *of food consumption.*

*Food-item recurrence strength,*$\alpha_{u,o}$: *The* food-item recurrence strength, *$\alpha_{u,o}$ for individual u during meal occasion o is the fraction of days in the food consumption sequence ${\left[m_{u,o,t}\right]}_{t=1}^{t=n_{u}}$ where at least one recurrent food-item was consumed by individual in the meal. The food-item recurrence strength indicates an individual’s tendency to eat at least one recurrent food-item in a given meal occasion. The food-item recurrence strength is computed as*
(3)$$\alpha_{u,o}=\sum_{m\in M_{u,o}}\frac{f_{u,o}\left(m\right)}{n_{u}}𝟙[|m\cap {\tilde{D}}_{u,o}\left|\geq 1\right].$$*Food-item-per-meal recurrence strength,*$\beta_{u,o}$: *The food-item-per-meal recurrence strength $\beta_{u,o}$ for individual u at meal occasion o is the average proportion of food-items in a meal that are recurrent food-items in their food consumption sequence ${\left[m_{u,o,t}\right]}_{t=1}^{t=n_{u}}$. The food-item-per-meal recurrence strength signifies an individual’s tendency to eat recurrent food-items in a given meal occasion. The food-item-per-meal recurrence strength is computed as*(4)$$\beta_{u,o}=\sum_{m\in M_{u,o}}\frac{f_{u,o}\left(m_{i}\right)}{n_{u}}\frac{|m\cap {\tilde{D}}_{u,o}|}{\left|m\right|}.$$*Meal recurrence strength,*$\gamma_{u,o}$: *The*meal recurrence strength*$\gamma_{u,o}$ for individual u at meal occasion o is the fraction of days in the food consumption sequence ${\left[m_{u,o,t}\right]}_{t=1}^{t=n_{u}}$ the individual consumed a recurrent meal. The meal recurrence strength signifies an individual’s tendency to consume a recurrent meal in a given occasion. The meal recurrence strength is computed as*(5)$$\gamma_{u,o}=\sum_{m\in M_{u,o}}\frac{f_{u,o}\left(m\right)\;𝟙\left[m\in {\tilde{M}}_{u,o}\right]}{n_{u}}.$$

**Example** **1.**
*Consider the example to understand the recurrent consumption computation. In the example shown in Table 2, recurrent food-items are latte and muffin because they meet both the frequency and span criterion. Similarly the recurrent meals are latte combined with muffin or latte by itself. The value of recurrent computation definitions for the example is presented in Table 2.*


## 3. Analysis Results

In this section, we present different patterns of recurrent consumption found by computing the recurrence strength tuple from the food logging data of MyFitnessPal users. First, we found the minimum number of days to reliable estimate the recurrent food-item set, recurrent meal set, and the recurrence strength tuple. Second, we studied the trends in the three recurrence strength measures that form the recurrence strength tuple. Third, we investigated the meal occasion differences and weekday/weekend differences in recurrence strength tuple. Lastly, we evaluated the association between recurrent consumption and the duration of logging. In subsequent sections, *p* denotes the *p*-value, and σ denotes the standard deviation.

### 3.1. Duration for Measurement of Recurrent Consumption Behavior

In this section, we determined the minimum number of days required to estimate an individuals’ recurrent consumption reliably. In the computation of recurrent consumption, we estimated five measures namely, recurrent food-item, recurrent meal set and the recurrence strength tuple $\left(\alpha_{u,o},\beta_{u,o},\gamma_{u,o}\right)$. For any individual, the quality of the estimated measures depends on the amount of food consumption data available for that individual. The amount of food consumption data translates to the length of the food consumption sequence in our analysis. We expect that the estimation error of the measures will be lower for long consumption sequences (more data) than for short consumption sequences (less data).

Hence, we inspected the dependence between the estimation error and the length of the consumption sequence quantitatively. To analyze the relationship, we considered a subset of individuals, n=203 individuals, who had long consumption sequences $n_{u}\geq 112$ days. First, we estimated the recurrent measures recurrent food-item set ${\tilde{D}}_{u,o,112}$, recurrent meal set ${\tilde{M}}_{u,o,112}$ and recurrence strength tuple $\left(\alpha_{u,o,112},\beta_{u,o,112},\gamma_{u,o,112}\right)$ from the 112 days long sequence as the baseline estimates. Next, we calculated the estimates with a shorter sequence length $\hat{n}$.

The estimates are denoted as ${\tilde{D}}_{u,o,\hat{n}}$, ${\tilde{M}}_{u,o,\hat{n}}$ , $\left(\alpha_{u,o,\hat{n}},\beta_{u,o,\hat{n}},\gamma_{u,o,\hat{n}}\right)$. To understand the estimation error, we compare the shorter sequence length estimates with the baseline estimates (long sequence estimate). For comparison, we constructed the estimation error definition to find the combined error in all the five recurrent consumption measures. The estimation error for a shorter sequence length $\hat{n}$, denoted as $e_{u,o}\left(\hat{n}\right)$, is presented below.
$$\begin{array}{cc} e_{u,o}\left(\hat{n}\right) & =\left(1-\frac{{\tilde{D}}_{u,o,\hat{n}}\cap {\tilde{D}}_{u,o,112}}{{\tilde{D}}_{u,o,\hat{n}}\cup {\tilde{D}}_{u,o,112}}\right)\left(\frac{|\alpha_{u,o\hat{n}}-\alpha_{u,o,112}|}{3}\right)\\ \; & +\left(1-\frac{{\tilde{D}}_{u,o,\hat{n}}\cap {\tilde{D}}_{u,o,112}}{{\tilde{D}}_{u,o,\hat{n}}\cup {\tilde{D}}_{u,o,112}}\right)\left(\frac{|\beta_{u,o,\hat{n}}-\beta_{u,o,112}|}{3}\right)\\ \; & +\left(1-\frac{{\tilde{M}}_{u,o,\hat{n}}\cap {\tilde{M}}_{u,o,112}}{{\tilde{M}}_{u,o,\hat{n}}\cup {\tilde{M}}_{u,o,112}}\right)\left(\frac{|\gamma_{u,o,\hat{n}}-\gamma_{u,o,112}|}{3}\right) \end{array}$$

We measured the estimation error in both the detection of recurrent choices (food-item or meal) and degree of consumption of those recurrent choices (recurrence strength) to determine the minimum sequence length needed to estimate recurrent consumption reliably. Hence, the estimation error consists of three parts with each part being a multiplication of two components. The first component measures the error in identifying recurrent choices, and the second component measures the absolute error in the recurrence strength.

The three parts correspond to the error in the three recurrence strength levels $\left(\alpha_{u,o},\beta_{u,o},\gamma_{u,o}\right)$. The error in the recurrent food-item set or recurrent meal set is calculated by measuring the Jaccard distance [25,26] between sets obtained with the shorter sequence length and the baseline set. The Jaccard distance is a distance measure on sets widely used in applications like data mining and information retrieval [25,26].

We computed the estimation error $e_{u,o}\left(\hat{n}\right)$ between the baseline estimates and estimates obtained for every $\hat{n}$ ranging from 14 days to 98 days. The estimation error can takes values ranging from zero to one. Figure 2b presents the observed trend of decrease in estimation error with increase in $\hat{n}$. We observe that the length of the sequence is $\hat{n}\geq 28$ provides an average estimation error of $e_{u,o}\leq 0.1$ (no units because the metrics are a ratio) across individuals. Figure 2a presents the Kaplan–Meier survival analysis curve [27] for the food consumption sequence length $n_{u}$ across the 2758 individuals. We observe that for approximately 80% of individuals, the sequence length is greater than 28 days.


*Thus, in the sequel, we only consider individuals with $n_{u}\geq 28$ in our analysis as data of 28 days or more provides reliable estimates of recurrent consumption. Hence, the results presented in the sequel are obtained with data from 1581 individuals as they have $n_{u}\geq 28$.*


### 3.2. Trends in Recurrent Consumption Behavior

This section presents an analysis of the recurrent food-items and the recurrence strength tuple computed for individuals in the dataset. As text descriptions represent food choices, Table 3 displays the top 50 words found in the food-items present in the recurrent food-item set across individuals in the dataset $\cup_{u}{\tilde{D}}_{u,o}$. Prior to creating the list, we filtered the food-item text descriptions with a food taxonomy [28] list to retain actual food words in the text description for the purposes of a more lucid list.

We found that *coffee, milk* is a commonly occurring word in habitual breakfast food-items across individuals in the dataset. For lunch and dinner, we found words like *chicken, salad* to be found in habitual food-items the dataset. For snacks, we found words like *apple, banana* to be among the common habitual food-item words. While the words themselves are not surprising, it does provide evidence that the framework detected habitual choices in the dataset reliably.

The recurrence strength tuple quantifies different levels of recurrent consumption behavior. Figure 3a indicates that 93% of individuals exhibit a non zero food-item recurrence strength $\alpha_{u,o}$ for breakfast, 65% for lunch, 58% for dinner, and 72% for snacks. Thus, individuals are likely to exhibit recurrent consumption behavior for food-items, i.e., they are likely to consume at least one recurrent food-item in their meal.

All food-items consumed in a meal may not be recurrent food-items because of individuals’ novelty-seeking behaviors. Thus, food-item-per-meal recurrence strength $\beta_{u,o}$, i.e., the average fraction of recurrent food-items consumed per meal is not as high as the food-item recurrence strength $\alpha_{u,o}$. Consequently, the cumulative distribution function (CDF) curve for food-item-per-meal recurrence strength is above the recurrent food-item curve in Figure 3a across all meal occasions.

Figure 3a indicates that a tiny fraction of individuals consumes recurrent meals regularly. Figure 3a shows that the fraction of individuals with non-zero meal recurrence strength $\gamma_{u,o}$ is 27% for breakfast and 6%, 3%, and, 7% for lunch, dinner and snacks, respectively. The meal recurrence strength is not high even for individuals with high food-item recurrence strength and high food-item-per-meal recurrence strength. The reason is that individuals, while choosing a meal, combine food-items in diverse and complex ways. Even though certain food-items are repeated daily (high food-item and high food-item-per-meal recurrence strength), they are not combined in the same combination each time, leading to different meals and a low meal recurrence strength.

Figure 3b shows a heat-map of the Pearson’s correlation coefficients between the recurrence strength tuple across meal occasions. We observed high correlations between food-item recurrence strength and food-item-per-meal recurrence strength for each meal occasion separately. We also observed moderate correlations between meal occasions implying that individuals with relatively high recurrence strength at a meal occasion likely have high recurrence strength at other occasions as well.

As each meal is a combination of food-items, the number of food-items selected in a meal is also a variable that may influence the recurrence strength tuple. The average number of food-item-per-meal is calculated as $\sum_{m\in M_{u,o}}\frac{\left|m\right|f_{u,o}\left(m\right)}{n_{u}}$. To test the influence of the number of food-items per meal on the recurrence strength tuple, we computed the Pearson’s correlation coefficient between recurrence strength tuple $\alpha_{u,o},\beta_{u,o},\gamma_{u,o}$ and the average number of food-items selected in a meal.

Table 4, Table 5, Table 6 and Table 7 presents the correlation between recurrence strength tuple and the average number of food-item-per-meal. The food-item recurrence strength and food-item-per-meal recurrence strength are positively correlated with the average number of food-item-per-meal. A high average number of food-items consumed in a meal leads to a high tendency to repeat food-item choices, which manifests as high consumption of recurrent food-items. However, the meal recurrence strength is negatively correlated with the average number of food-item-per-meal, suggesting that a high average number of food-item choices per meal leads to an increase in the possibilities of food-item choice combinations and a decrease in repetitions of the exact meal.


*We observe that individuals’ have a higher tendency of consuming habitual food-items in their meal in different combinations than a habitual meal itself. Individuals who consume a high number of food-items per meal on average tend to consume habitual food-items in their meal but not a habitual meal. Individuals with a low average number of food-items per meal are more likely to consume a habitual meal.*


### 3.3. Role of Meal Occasions in Recurrent Consumption Behavior

In this section, we investigated the difference in recurrent consumption behavior across meal occasions. Although the trend in recurrence strength tuples was consistent across meal occasions, we found the actual values differed considerably across meal occasions. We used the Friedman statistical test to investigate differences between the recurrence strength tuple across meal occasions for the same group of individuals. We used the Wilcoxon signed-rank test for pairwise post hoc comparisons.

The food-item recurrence strength is significantly (p<0.01) different for all meal occasion pairs. The food-item recurrence strength $\alpha_{u,o}$ is the highest for breakfast followed by snacks, lunch and dinner. The food-item-per-meal recurrence strength is also significantly (p<0.01) different for all meal occasion pairs. The food-item-per-meal recurrence strength $\beta_{u,o}$ follows the same trend—that is, breakfast is the highest. Next are snacks, lunch, and dinner in decreasing order.

The meal recurrence strength is statistically different (p<0.01) across all meal occasion pairs except lunch and snacks (p=0.343). The meal recurrence strength $\gamma_{u,o}$ is highest for breakfast. Followed by snacks, lunch, and dinner. The comparisons between meal occasions for recurrence strength tuple are displayed in Figure 4 and statistics are presented in Table 8. The findings are consistent with previous work [5] that found a similar trend in the fraction of daily repeat consumption, i.e., the fraction of items in a meal that was consumed in the previous seven days.


*We observed that individuals showed the highest habitual consumption for breakfast for both food-items and meals, followed by snacks, and then lunch. Individuals exhibited the least habitual consumption at dinner. For individual food-items, habitual behavior at breakfast was 2–3X more than at other meal occasions. However, at meals, the habitual behavior at breakfast was 10X more than at other meal occasions.*


### 3.4. Role of Weekday/Weekend Pattern in Recurrent Consumption Behavior

We investigated the effect of the weekday–weekend context on the recurrence strength tuple. We studied if the consumption of habitual food-items was different between weekdays and weekends. In the food consumption sequence, we separated the weekday days from the weekend days. We computed the recurrent consumption behavior separately for the weekday consumption sequence and weekend consumption sequence.

The number of weekend days (length of the weekend food consumption sequence) might be very low for users as logging activity during the weekend tends to be lower than weekday logging activity [5]. To minimize erroneous estimates due to a low number of weekend days, we consider individuals with several weekend days ≥14 days. Hence, for the weekday–weekend analysis presented in Figure 5, the number of individuals is 686.

We used the Wilcoxon signed ranked test to evaluate the difference between weekday food-item-per-meal recurrence strength and weekend food-item-per-meal recurrence strength as the individuals were the same across both groups. We find that the food-item-per-meal recurrence strength is significantly greater on weekdays than weekends across breakfast ($p=2\times 10^{-27}$), lunch ($p=1\times 10^{-19}$), dinner (p=0.002) and snacks ($p=7\times 10^{-13}$). The difference is largest for breakfast with a mean of 0.063 (σ = 0.14). Followed by lunch with a mean of 0.055 (σ = 0.15), snacks with a mean of 0.036 (σ = 0.13) and dinner with a mean of 0.01 (σ = 0.1).

We investigated the associations between weekday and weekend food-item-per-meal recurrence strength using Pearson’s correlation coefficient. The correlation coefficient (significance) is 0.84 ($p=7\times 10^{-184}$) for breakfast, 0.71 ($p=3\times 10^{-107}$) for lunch, 0.81 ($p=4\times 10^{-164}$) for dinner and 0.80 ($p=2\times 10^{-151}$) for snacks. Such a high correlation suggests that individuals with relatively high recurrent consumption behavior on weekdays also exhibit relatively high recurrent consumption behavior on weekends.


*We observed that the consumption of recurrent food-items was higher on weekdays compared with on weekends. Furthermore, the consumption of recurrent food-items on weekdays was correlated with recurrent food-item consumption on weekends. The correlation suggests that individuals with a relatively high correlation on weekdays will also have a relatively high correlation on weekends even though their weekend consumption is lower than their weekday consumption.*


### 3.5. Association with Food Logging

The individuals studied in this work are MyFitnessPal users from a weight-loss community. There is a possibility of influence of MyFitnessPal usage behavior on recurrent consumption. In this section, we investigated the association between recurrence strength and the number of recorded days of an individual. The number of recorded days is the number of days where the user has entered at least one meal occasion. Note that we use the number of recorded days as a proxy measurement for engagement in food logging.

Table 4, Table 5, Table 6 and Table 7 indicates an statistically insignificant correlation between the recurrence strength and the number of recorded days for different meal occasions. *Hence, we did not find a statically significant association between engagement in logging and recurrent consumption behavior.*

## 4. Discussion and Conclusions

Previous research [5,15] built algorithms that used past diet information to learn individual’s food preferences and provide personalized food recommendations. We analyzed the past diet information of individuals to build an intuition about their habitual choices and the extent of recurrence of those choices, which together, reveal information about the food habit forming capability of the individuals. We introduced a framework to identify recurrent food choices from individuals’ food diary data and estimate their habit-forming capability in three levels, i.e., food-item recurrence strength, food-item-per-meal recurrence strength, and meal recurrence strength. Based on the key insights revealed in the analysis of these measures in the population of MyFitnessPal users, we identify certain opportunities to improve diet self management strategies.

**Diet behavior profile of individuals using self-monitoring diet apps**: Diet management strategies have been primary focused on nutrition. Personalization based on individual’s dietary behaviors will make management strategies easier to adopt and sustain. The computational measures proposed in this work can be used to build a diet behavior profile of an individual. First, data from the food diary may be used to identify specific habitual food choices (recurrent food choices) for replacement if unhealthy. If the habitual choices are healthy, they may be used to suggest the replacement of unhealthy choices. The suggestion of healthy habitual choices may be easier to adopt as the individual is familiar with it and can easily incorporate it into their routine.

Secondly, an individual’s habit-forming capacity should be measured with the recurrence strength tuple at all three levels, and the level with the highest recurrent consumption behavior could be targeted. For example, for individuals who exhibit high food-item recurrence strength but low food-item-per-meal or meal recurrence strength, an intervention designed around specific food-items may work best. For individuals with high meal recurrence strength, a meal can be targeted for modification.

**Consideration of meal occasion variability**: From our analysis, it is clear that food consumption behavior differs across meal occasions. Hence, food interventions for each meal occasion should be designed separately. Recurrent meals can be targeted for breakfast, while a couple of food-items in a meal or a single food-item could be targeted for lunch and dinner, respectively. Food intervention for behavior change may focus on a single meal occasion to simplify the individual’s intervention.

**Consideration of weekend vs. weekday behaviors**: Food intervention suggestions should be closer to the individual’s food space for individuals with moderate recurrent consumption behavior on weekdays. On the contrary, suggestions could be a novel food-item or non-recurrent choice for weekends to support the novelty-seeking behavior of individuals. Individuals with very high recurrent consumption behavior on weekdays exhibit high recurrent consumption behavior on weekends as well relative to others.

Hence, food intervention for such individuals may be partially applicable to weekends as well. For individuals with low recurrent consumption behaviors, suggesting novel food items may be more effective. Twenty-eight days is not sufficient to observe weekend recurrent consumption behavior because there are only eight weekend days of data. However, measurement of weekday behavior can still tell a story for weekend behaviors as the recurrent consumption measures between weekdays and weekends were highly correlated.

**A month of data to measure habitual consumption**: From our analysis, we found that 28 days of a food diary can be used to identify and quantify recurrent consumption behavior for the next four months with an average estimation error of less than 0.1 (no units). For breakfast, the average estimation error was less than 0.1 for even 14 days. Thus, for applications where only breakfast monitoring is key, data collection of breakfast records for a short duration of 14 days is sufficient to study breakfast habitual consumption behavior. However, for other meal occasions, a longer duration of food logging may be required.

**Challenge in self-selected user data**: One of the challenges of a public food diary dataset is that we cannot validate the existence of the users. However, we feel that our choices of including and thus excluding some users could safeguard us from such issues. Note that food logging requires frequent user engagement with the app, and has no immediate rewards. Thus, by including consistent users in our analysis, who logged all four meal occasions for at least 28 days, we believe that the consistent users are real users who used the app for their personal fitness goals.

The individuals in the MyFitnessPal dataset may be from a limited sample of the population who are health conscious. However, previous work [18] found that individuals in the dataset do not necessarily follow a healthy diet but that measurement of dietary behaviors from such a population still provides meaningful insights. Thus, although our analysis is restricted to MyFitnessPal users, our proposed computational framework, reported values, and observed trends could guide other researchers to repeat the analysis in different populations using different diet diarization methods.

**Challenge in self-reported user data**: MyFitnessPal users are not trained to complete food diaries accurately as in established nutrition studies, and hence the data suffers from potential inaccuracies and omissions. We addressed the challenge by extensive and careful data preprocessing. We reduced the impact of omission in our analysis by considering the days when all four meal occasions were recorded. The confidence in our analysis is enhanced by the fact that our statistical insights overlap with the common notions of habitual behavior. Finally, we note that self-selected self-reported food diaries are here to stay in some form or the other (at least in the foreseeable future), and thus it is important we investigate and analyze them.

**Complements current diet methodologies**: Food and agriculture are among the major driving systems of the Global Syndemic of Obesity, Undernutrition, and Climate change. It is all the more critical to design strategies that effectively push the eating behavior towards sustainable diets that are healthy with a low carbon footprint. We fathom that our methodology of computing recurrent food items and meals can complement preexisting methods that characterize diet quality quantitatively. Food databases [29] can be used to identify the higher-level food groups of the recurrent food items and quantify adherence scores to various diet recommendations, such as the Mediterranean diet [22] or the one-health approach Cultural Double Pyramid [21].

Identifying recurrent food items may also lead to identifying sustainable local food items that can be encouraged and ultra-processed foods items [30] that need to be replaced. Many current recommendations, such as Healthy Eating Plate [31] or Myplate [32] provide one-size-fits-all guidelines. The proposed methodology and derived insights can advance food recommendation AI to provide more sustainable and personalized food suggestions learned from their recurrent food patterns. While beyond the scope of this paper, we note that the privacy of user-contributed data is an important issue in diet-tracking apps, such as MyFitnessPal [13].

**Limitations and future work**: We did not find statistically significant demographic differences in the recurrence strength tuple contrary to Liu et al. A reason could be the reasonably skewed demographic distribution in the dataset. Our current sample is predominantly female and of the age group between 18 and 44 years. Our findings may have limited generalization. Hence, a future direction is to collect diet diary datasets from diverse populations to investigate demographic differences in recurrent consumption behavior.

An outstanding example is a recent study [33] that analyzed a large diet tracking dataset and observed that food environment, i.e., higher access to grocery stores, lower access to fast food, higher income and college education were associated with healthier food consumption. However, the associations differed across locations with predominantly Black, Hispanic, or white populations. Future work could also investigate how the demographic and socio-cultural factors impact dietary patterns, as measured by our proposed computational framework.

Along with quantifying an individual’s behavior, it is imperative to understand the drivers behind such behavior. While we reported the trends and values of recurrent consumption behavior, we did not have access to any information to investigate the reasons behind high or low recurrent consumption behavior. High recurrent consumption behavior could manifest due to numerous reasons, including the individual’s food insecurity or personality or access to limited options in the workplace. We will explore connections between socio-cultural factors, personality, appetite traits, food choice determinants, and recurrent consumption behavior in the future. Towards that end, we are currently conducting a study where we plan to collect more recent food diary data along with demographic, personality, and eating behavior questionnaires [34].

In our analysis, we modeled each food-item independently. However, individuals may pair or group certain food-items and repeat the group frequently even though the entire meal is not repeated precisely. Thus, a future direction would be to identify groups of recurrent food-items on an individual or population level.

**Conclusion:** Recent studies have analyzed large datasets collected through self-monitoring mobile health applications [33,35,36] to derive insights about human behavior. While these datasets may suffer from inaccuracies due to their self-reporting nature, they still provide a valuable resource for studying long-term human behavior in free-living conditions. In this work, we developed and analyzed computational measures of habitual food consumption behavior from food diaries. We introduced a new behavioral measure for habitual food consumption, namely recurrent consumption.

We developed computational measures to identify and quantify the recurrence strength for food-items and meals automatically from food diaries. We discussed the types of recurrent consumption behaviors and their dependence on temporal contextual factors elaborately with the help of a large public food diary dataset. The proposed method provides quantifiable measures to enable new opportunities for personalizing food intervention designs based on the person’s habitual behavior profile.

## Figures and Tables

**Figure 1 sensors-22-02753-f001:**
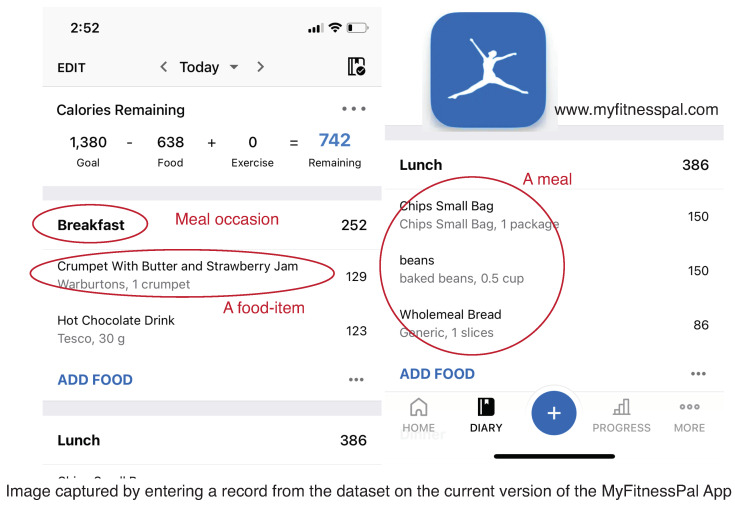
Example of MyFitnessPal App’s interface [13].

**Figure 2 sensors-22-02753-f002:**
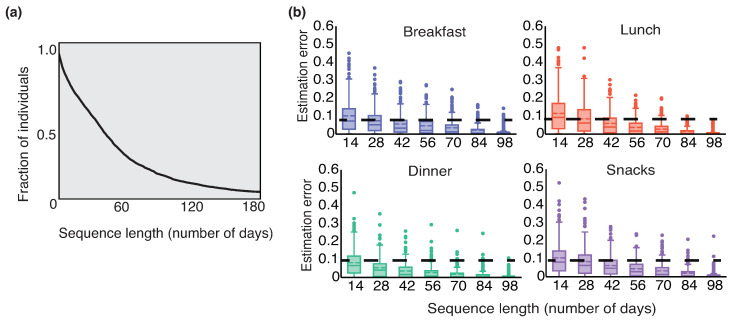
(**a**) Kaplan–Meier estimated survival analysis for the number of complete recorded days. (**b**) Decreasing trend in estimation error as the consumption sequence length (number of recorded days) increases. We chose 28 days for analysis as this allows less than 0.1 estimation error in recurrent consumption.

**Figure 3 sensors-22-02753-f003:**
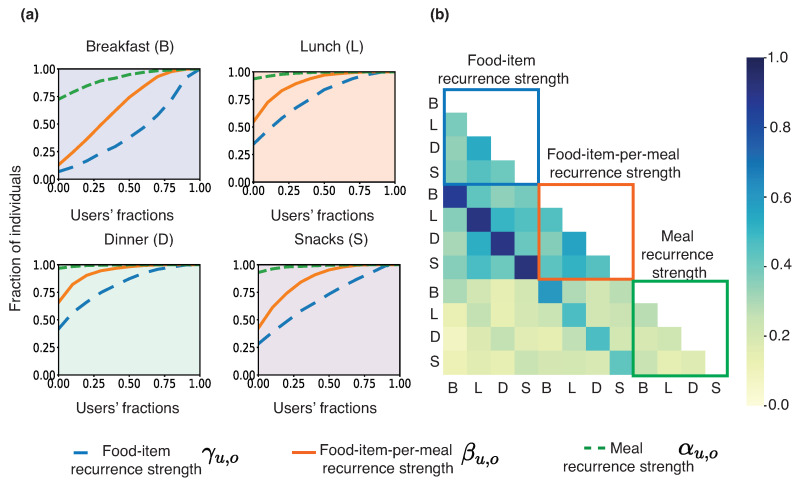
(**a**) The cumulative distribution function of the recurrence strength tuple across 1581 individuals. (**b**) Heatmap of correlations between the recurrence strength tuple across all meal occasions.

**Figure 4 sensors-22-02753-f004:**
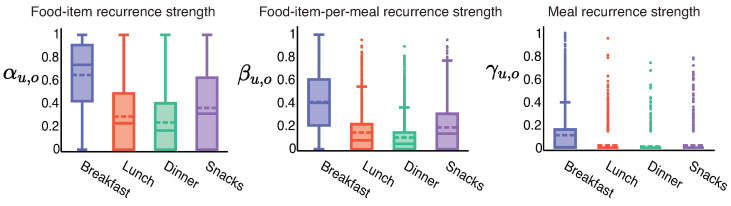
Differences in recurrence strength tuple across meal occasions. The number of individuals in each box plot is 1581.

**Figure 5 sensors-22-02753-f005:**
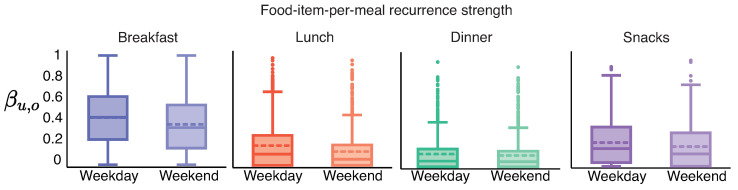
Differences in recurrence strength tuple between weekdays and weekends for breakfast, lunch, dinner, and snacks. The number of individuals in every box plot is 686.

**Table 1 sensors-22-02753-t001:** Demographic description of the users in the MyFitnessPal Public Dataset.

	Number of Users	Gender			Age Group			Region				
		F	M	U	18–44	45+	U	NE	MW	S	W	U
All users in dataset	9896	73%	16%	11%	71%	18%	11%	12%	16%	21%	14%	37%
Final analyzed users	1581	73%	15%	12%	71%	18%	11%	11%	16%	21%	13%	39%

F: Female, M: Male, U: Missing, NE: Northeast, MW: Midwest, S: South, W: West.

**Table 2 sensors-22-02753-t002:** Computed values for the example.

Framework Variables	Value
Individual identifier *u*	1
Meal occasion *o*	breakfast
duration $n_{u}$	14
Meal consumption sequence ${\left[m_{u,o}\right]}_{t=1}^{t=n_{u}}$	[{latte, muffin}, {latte}, {latte, muffin}, {latte, croissant}, {latte}, {latte}, {latte, muffin}, {latte, muffin}, {latte}, {latte, muffin}, {latte, muffin}, {muffin, hot-chocolate}, {latte}, {latte, muffin}]
Food-item library $D_{u,o}$	{latte, croissant, muffin, hot-chocolate}
Meal library $M_{u,o}$	{{latte, muffin}, {latte, croissant}, {latte}, {muffin, hot-chocolate} }
Recurrent food-item set ${\tilde{D}}_{u,o}$	{ latte, muffin }
Recurrent meal set ${\tilde{M}}_{u,o}$	{{latte, muffin}, {latte}}
Food-item recurrence strength $\alpha_{u,o}$	$\frac{1+1+1+1+1+1+1+1+1+1+1+1+1+1}{14}=1$
Food-item-per-meal recurrence strength $\beta_{u,o}$	$\frac{7(1)+1(0.5)+5(1)+1(0.5)}{14}=0.93$
Meal recurrence strength $\gamma_{u,o}$	$\frac{7+0+5+0}{14}=0.86$
Recurrence strength tuple $\left(\alpha_{u,o},\beta_{u,o},\gamma_{u,o}\right)$	(1,0.93,0.86)

**Table 3 sensors-22-02753-t003:** The top 50 tokens found in text created by combining recurrent food-item sets across individuals $\cup_{u}{\tilde{D}}_{u,o}$. The table presents the common words that are present in habitual food-items in the dataset.

breakfast	milk, coffee, egg, sugar, butter, almond, banana, vanilla, protein, whole, chocolate, creamer, fat, cream, and, yogurt, bread, oats, cheese, oatmeal, skimmed, peanut, bananas, wheat, fruit, great, brown, honey, oil, white, eggs, bacon, fresh, cereal, tea, whey, spinach, french, cinnamon, blueberries, shake, coconut, bar, powder, grain, liquid, fried, frozen, strawberry, reduced
lunch	chicken, cheese, salad, bread, lettuce, raw, fresh, cucumber, sweet, dressing, turkey, whole, butter, spinach, fat, oil, wheat, baby, rice, white, tomato, olive, ham, and, green, apple, yogurt, milk, red, tuna, cheddar, in, pepper, chocolate, roasted, deli, mix, broccoli, grilled, protein, sliced, sandwich, egg, brown, grain, boiled, honey, cherry, hard, peppers
dinner	chicken, cheese, oil, butter, rice, sweet, salad, broccoli, olive, fresh, green, white, raw, bread, potato, lettuce, spinach, beans, dressing, cooked, red, fat, cheddar, and, extra, steamed, tomato, milk, frozen, whole, sauce, grilled, wine, baby, cream, cucumber, great, onion, salted, brown, vegetable, garden, baked, turkey, peppers, virgin, potatoes, peas, sour, steak
snacks	chocolate, milk, protein, butter, bar, apple, banana, peanut, cheese, cream, coffee, almonds, yogurt, fat, vanilla, raw, sugar, almond, dark, and, popcorn, fruit, nuts, whey, bananas, apples, skimmed, tea, whole, honey, red, roasted, fresh, cookies, ice, pop, coconut, chip, white, salt, great, chips, mini, orange, powder, creamy, strawberry, rice, mix, salted

**Table 4 sensors-22-02753-t004:** Pearson’s correlation coefficient ρ and significance indicated by the *p*-value between recurrence strength tuple and other variables for breakfast.

Breakfast				
**Recurrence Strength**	**Average Number of Food-Items per Meal**		**Number of Recorded Days**	
	ρ	***p*-Value**	ρ	***p*-Value**
**Food-item** recurrence strength	0.47	5 $\times \;10^{-89}$	0.04	0.09
**Food-item-per-meal** recurrence strength	0.27	7 $\times \;10^{-28}$	0.02	0.49
**Meal** recurrence strength	−0.25	6 $\times \;10^{-24}$	0.002	0.91

**Table 5 sensors-22-02753-t005:** Pearson’s correlation coefficient ρ and significance indicated by the *p*-value between recurrence strength tuple and other variables for **lunch**.

Lunch				
**Recurrence Strength**	**Average Number of Food-Items per Meal**		**Number of Recorded Days**	
	ρ	***p*-Value**	ρ	***p*-Value**
**Food-item** recurrence strength	0.50	7 $\times \;10^{-102}$	−0.01	0.82
**Food-item-per-meal** recurrence strength	0.34	6 $\times \;10^{-44}$	0.02	0.35
**Meal** recurrence strength	−0.14	2 $\times \;10^{-8}$	0.05	0.05

**Table 6 sensors-22-02753-t006:** Pearson’s correlation coefficient ρ and significance indicated by the *p*-value between recurrence strength tuple and other variables for **dinner**.

Dinner				
**Recurrence Strength**	**Average Number of Food-Items per Meal**		**Number of Recorded Days**	
	ρ	***p*-Value**	ρ	***p*-Value**
**Food-item** recurrence strength	0.47	1 $\times \;10^{-86}$	0.02	0.41
**Food-item-per-meal** recurrence strength	0.28	8 $\times \;10^{-31}$	0.03	0.27
**Meal** recurrence strength	−0.09	5 $\times \;10^{-4}$	0.05	0.07

**Table 7 sensors-22-02753-t007:** Pearson’s correlation coefficient ρ and significance indicated by the *p*-value between recurrence strength tuple and other variables for **snacks**.

Snacks				
**Recurrence Strength**	**Average Number of Food-Items per Meal**		**Number of Recorded Days**	
	ρ	***p*-Value**	ρ	***p*-Value**
**Food-item** recurrence strength	0.49	6 $\times \;10^{-95}$	−0.02	0.43
**Food-item-per-meal** recurrence strength	0.27	1 $\times \;10^{-29}$	−0.01	0.62
**Meal** recurrence strength	−0.21	1 $\times \;10^{-16}$	0.02	0.42

**Table 8 sensors-22-02753-t008:** The mean (standard deviation) statistics of recurrence strength measures across meal occasions.

Recurrence Strength	Breakfast	Lunch	Dinner	Snacks
**Food-item** recurrence strength	0.65(0.30)	0.28(0.28)	0.24(0.27)	0.36(0.32)
**Food-item-per-meal** recurrence strength	0.42(0.25)	0.15(0.18)	0.10(0.15)	0.19(0.20)
**Meal** recurrence strength	0.11(0.21)	0.02(0.08)	0.01(0.05)	0.02(0.08)

## Data Availability

The original raw dataset [16] is publicly available at [37]. The custom code developed for this study is available from the corresponding author on reasonable request.

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
