# Peer review of "Food Habits: Insights from Food Diaries via Computational Recurrence Measures"

_sensors, 2022, doi:10.3390/s22072753_

Round 1
Reviewer 1 Report
Congrats! to the authors
This is very well-written and well-organized paper, I recommend to accept in present form.
Author Response
Response to Reviewer 1:
Congrats! to the authors
This is very well-written and well-organized paper, I recommend to accept in present form.
We thank the reviewer for their time and effort in reviewing our manuscript.
Reviewer 2 Report
Authors could improve the research with more recent data from the applications for a future manuscript.
Author Response
Response to Reviewer 2:
We thank the reviewer for their time and effort in reviewing our manuscript.
Authors could improve the research with more recent data from the applications for a future manuscript.
We agree with the reviewer. We are currently conducting a study to collect more recent food diaries with demographic, personality, and eating behavior data. We added a line discussing it in the manuscript line 621.
Reviewer 3 Report
- the work refers to a methodology that may complement that of questionnaires of dietary scores (to quantify the adherence to a given dietary model as the Mediterranean Diet (more info for ex. here: Zaragoza-Martí A, Cabañero-Martínez MJ, Hurtado-Sánchez JA, Laguna-Pérez A, Ferrer-Cascales R. Evaluation of Mediterranean diet adherence scores: a systematic review. BMJ Open. 2018 Feb 24;8(2):e019033. doi: 10.1136/bmjopen-2017-019033.). Authors are advised to acknowledge other methodologies used to characterize food habits
- Authors assume that food choices follow the same trend as any other purchasing choice, which is an interesting view to characterise current dietary transition towards a uniform global unsustainable diet vs sustainable food habits based on local foods and avoidance of ultraprocessed foods and fast foods (authors may wish to enrich introduction and discussion of results with relevant information on dietary habits such as:
Barilla Foundation & Research Unit on Nutrition, Diabetes and Metabolism, University of Naples Federico II, 2021. A one health approach to food, the Double Pyramid connecting food culture, health and climate.
Swinburn BA, Kraak VI, Allender S, Atkins VJ, Baker PI, Bogard JR, Brinsden H, Calvillo A, De Schutter O, Devarajan R, Ezzati M, Friel S, Goenka S, Hammond RA, Hastings G, Hawkes C, Herrero M, Hovmand PS, Howden M, Jaacks LM, Kapetanaki AB, Kasman M, Kuhnlein HV, Kumanyika SK, Larijani B, Lobstein T, Long MW, Matsudo VKR, Mills SDH, Morgan G, Morshed A, Nece PM, Pan A, Patterson DW, Sacks G, Shekar M, Simmons GL, Smit W, Tootee A, Vandevijvere S, Waterlander WE, Wolfenden L, Dietz WH. The Global Syndemic of Obesity, Undernutrition, and Climate Change: The Lancet Commission report. Lancet. 2019 Feb 23;393(10173):791-846. doi: 10.1016/S0140-6736(18)32822-8.
Delgado A, Issaoui M, Vieira MC, Saraiva de Carvalho I, Fardet A. Food Composition Databases: Does It Matter to Human Health? Nutrients. 2021 Aug 17;13(8):2816. doi: 10.3390/nu13082816.
Elizabeth L, Machado P, Zinöcker M, Baker P, Lawrence M. Ultra-Processed Foods and Health Outcomes: A Narrative Review. Nutrients. 2020 Jun 30;12(7):1955. doi: 10.3390/nu12071955.
- As referred above, authors are advised to address classical methodologies (diet adherence scores based on pre-defined anonymised queries) and to explain the compliance of their methodology to best practices on personal data protection (having in mind strong law enforcement in protecting personal data, for example, in Europe) especially when linked to commercial exploitation of such personal data - e.g., customised suggestions to users based on their profiling;
- authors are advised to replace "dishes" (an apple can be a dish?) and meals ("coffee break" usually means snack time, if so, is black coffee, without sugar, a meal?); suggestion use "food item" for a single item and "meal" or "various food items", depending on what makes sense; please note that in traditional diets food habits are rooted in culture and history and linked to the landscape - meals are not just eating food items - the designations breakfast, lunch, tea, dinner should be coherent with meal types and pauses; please review the literature on healthy and sustainable diets, as well as recommendations, in addition (but inline) to those from Barilla foundation (ref above), such as: https://www.hsph.harvard.edu/nutritionsource/healthy-eating-plate/ (to forgetting FAO/WHO recommendations) to enrich your discussion; please adjust your ow definitions accordingly
- the suggested changes aim at highlighting the meaning and usefulness of the presented methodology; the "recurrent choices" and "recurrent strength" of food choices should be in coherence with the suggested changes in other definitions. The coherency and adequacy of definitions is especially relevant to the model, having in mind that data on quantities, nutrient and energy intake is not considered; The focus will be therefore at the level of the variety in the diet and on eating routines;
- more information on the sample population should be provided in discussion to contextualize results, otherwise results as those in table 2 are meaningless in the viewpoints of public health, environmental, nutrition, food security; the conclusions should stress methodological constraints while highlighting the advantages for future (ethical) uses; The extrapolation to different populations and prospective uses should be discussed in view of the diverse demographic and socio-cultural determinants as well as the linkage to climate
Author Response
Response to Reviewer 3:
We thank the reviewer for their time and effort in reviewing our manuscript.
- the work refers to a methodology that may complement that of questionnaires of dietary scores (to quantify the adherence to a given dietary model as the Mediterranean Diet (more info for ex. here: Zaragoza-Martí A, Cabañero-Martínez MJ, Hurtado-Sánchez JA, Laguna-Pérez A, Ferrer-Cascales R. Evaluation of Mediterranean diet adherence scores: a systematic review. BMJ Open. 2018 Feb 24;8(2):e019033. doi: 10.1136/bmjopen-2017-019033.). Authors are advised to acknowledge other methodologies used to characterize food habits
We have revised the introduction describing the current methodologies used to capture and characterize habitual food intake (lines 29-34 and lines 116-122).
- Authors assume that food choices follow the same trend as any other purchasing choice, which is an interesting view to characterise current dietary transition towards a uniform global unsustainable diet vs sustainable food habits based on local foods and avoidance of ultraprocessed foods and fast foods (authors may wish to enrich introduction and discussion of results with relevant information on dietary habits such as:
Barilla Foundation & Research Unit on Nutrition, Diabetes and Metabolism, University of Naples Federico II, 2021. A one health approach to food, the Double Pyramid connecting food culture, health and climate.
Swinburn BA, Kraak VI, Allender S, Atkins VJ, Baker PI, Bogard JR, Brinsden H, Calvillo A, De Schutter O, Devarajan R, Ezzati M, Friel S, Goenka S, Hammond RA, Hastings G, Hawkes C, Herrero M, Hovmand PS, Howden M, Jaacks LM, Kapetanaki AB, Kasman M, Kuhnlein HV, Kumanyika SK, Larijani B, Lobstein T, Long MW, Matsudo VKR, Mills SDH, Morgan G, Morshed A, Nece PM, Pan A, Patterson DW, Sacks G, Shekar M, Simmons GL, Smit W, Tootee A, Vandevijvere S, Waterlander WE, Wolfenden L, Dietz WH. The Global Syndemic of Obesity, Undernutrition, and Climate Change: The Lancet Commission report. Lancet. 2019 Feb 23;393(10173):791-846. doi: 10.1016/S0140-6736(18)32822-8.
Delgado A, Issaoui M, Vieira MC, Saraiva de Carvalho I, Fardet A. Food Composition Databases: Does It Matter to Human Health? Nutrients. 2021 Aug 17;13(8):2816. doi: 10.3390/nu13082816.
Elizabeth L, Machado P, Zinöcker M, Baker P, Lawrence M. Ultra-Processed Foods and Health Outcomes: A Narrative Review. Nutrients. 2020 Jun 30;12(7):1955. doi: 10.3390/nu12071955.
We have revised the manuscript based on the reviewer’s suggestion and included a paragraph in the discussion section (lines 585-599) describing the potential uses of our methodology with respect to the current literature on dietary habits and recommendations.
- As referred above, authors are advised to address classical methodologies (diet adherence scores based on pre-defined anonymised queries) and to explain the compliance of their methodology to best practices on personal data protection (having in mind strong law enforcement in protecting personal data, for example, in Europe) especially when linked to commercial exploitation of such personal data - e.g., customised suggestions to users based on their profiling;
The reviewer brings up an important point about user privacy. Our focus is extracting useful insights from user data. We hope our results are used to perform analyses with full disclosure and consent from the user. Beyond our hope of ethical use, only regulations and laws can keep companies in check, and thus, enforcing privacy lies beyond the scope of this paper.
- authors are advised to replace "dishes" (an apple can be a dish?) and meals ("coffee break" usually means snack time, if so, is black coffee, without sugar, a meal?); suggestion use "food item" for a single item and "meal" or "various food items", depending on what makes sense; please note that in traditional diets food habits are rooted in culture and history and linked to the landscape - meals are not just eating food items - the designations breakfast, lunch, tea, dinner should be coherent with meal types and pauses; please review the literature on healthy and sustainable diets, as well as recommendations, in addition (but inline) to those from Barilla foundation (ref above), such as: https://www.hsph.harvard.edu/nutritionsource/healthy-eating-plate/ (to forgetting FAO/WHO recommendations) to enrich your discussion; please adjust your ow definitions accordingly
We agree and have revised our terminology and changed “dishes” to “food-items” based on the suggestion from the reviewer.
- the suggested changes aim at highlighting the meaning and usefulness of the presented methodology; the "recurrent choices" and "recurrent strength" of food choices should be in coherence with the suggested changes in other definitions. The coherency and adequacy of definitions is especially relevant to the model, having in mind that data on quantities, nutrient and energy intake is not considered; The focus will be therefore at the level of the variety in the diet and on eating routines;
We have revised the manuscript and figures to reflect the changes in terminology.
- more information on the sample population should be provided in discussion to contextualize results, otherwise results as those in table 2 are meaningless in the viewpoints of public health, environmental, nutrition, food security; the conclusions should stress methodological constraints while highlighting the advantages for future (ethical) uses; The extrapolation to different populations and prospective uses should be discussed in view of the diverse demographic and socio-cultural determinants as well as the linkage to climate
In Table 1 we present the sample population statistics. We also discuss the characteristics of our sample population in the discussion section in line 602. We then discuss (line 602-622) future directions to conduct new studies to understand the links between demographic, socio-cultural factors to recurrent consumption behavior.
Reviewer 4 Report
- The overall structure of the article is very full, but I think the Literature Review part is missing. The previous research on habitual consumption in the Introduction chapter can be placed in the Literature Review chapter to illustrate the inspiration and innovation of past research on this study, and it is more appropriate to add some introductions to the background of the article's topic selection in the Introduction part.
- The number of references cited in the paper is slightly insufficient, especially the MyFitnessPal food diary dataset and method introduction part, which should be improved by adding relevant research appropriately.
- In the section describing the trend of the three recurrence strength measures forming the recurrence strength tuple using sliced boxplots, it is considered to use a heatmap to show correlations.
Author Response
We thank the reviewer for their time and effort in reviewing our manuscript.
The overall structure of the article is very full, but I think the Literature Review part is missing. The previous research on habitual consumption in the Introduction chapter can be placed in the Literature Review chapter to illustrate the inspiration and innovation of past research on this study, and it is more appropriate to add some introductions to the background of the article's topic selection in the Introduction part.
We appreciate the reviewer’s suggestions. Due to the limited revision time (6 days) available to us, we were unable to completely adopt the full suggestion. However, we have made a number of changes to the introduction and discussion sections, and we hope our changes essentially meet the suggestion.
The number of references cited in the paper is slightly insufficient, especially the MyFitnessPal food diary dataset and method introduction part, which should be improved by adding relevant research appropriately.
We have added the citation for the MyFitnessPal food diary dataset in the method introduction part. We have also added content in the introduction and discussion sections citing relevant papers.
In the section describing the trend of the three recurrence strength measures forming the recurrence strength tuple using sliced boxplots, it is considered to use a heatmap to show correlations.
We added a heatmap showing correlations between the recurrence strength tuple across all meal occasions. We have explained our observation in lines 420-428.
Reviewer 5 Report
The paper indicates an interesting gap in food recommendation AI. This gap meant recurrent food habits that should be incorporated into food recommendations systems. The authors defined measured for the food-meal-dish recurrence.
The paper is interesting. There are a few minor deficiencies:
- I would find the paper more convincing if there is an example of the used app interface and at least a one-user-day data record. Now I can see only a short sample (Table 1) and most frequent words (Table 2); however, I hesitate if words such as “fresh”, “hard”, “baked” a full meal or dish.
- I appreciate consistent and meaningful colouring of figures. I believe Figure 2 would be easier to understand if the X-axis indicates users' fractions.
- The app's name “MyFitnessPal" should be consistent throughout the article, e.g. see line 84.
- Paragraph 2.1. can be accompanied by a table with all the descriptive dataset statistics and an example (I mentioned before).
- I appreciate the notation and the measures. However, it isn't easy to read without an example, e.g., differentiating between dishes and meals.
- I do not understand why the criterion for the minimal span is 1/2 in equations (1) and (2).
- The idea of incorporating a span for measuring food recurrence is compelling.
- Often citations are not preceded by a space, e.g. see line 346.
- The text from line 418 is hard to read, and it is better to accompany it with a table. What is this indicator in parenthesis?
Author Response
Response to Reviewer 5:
We thank the reviewer for their time and effort in reviewing our manuscript.
The paper indicates an interesting gap in food recommendation AI. This gap meant recurrent food habits that should be incorporated into food recommendations systems. The authors defined measured for the food-meal-dish recurrence.
The paper is interesting. There are a few minor deficiencies:
- I would find the paper more convincing if there is an example of the used app interface and at least a one-user-day data record. Now I can see only a short sample (Table 1) and most frequent words (Table 2); however, I hesitate if words such as “fresh”, “hard”, “baked” a full meal or dish.
We have created an image with the illustration of the current version of the MyFitnessPal app in Figure 1. An example of one user’s record for a day is attached as a supplementary file1.
- I appreciate consistent and meaningful colouring of figures. I believe Figure 2 would be easier to understand if the X-axis indicates users' fractions.
The label for x-axis had been changed to users’ fractions as per the reviewer’s suggestion.
- The app's name “MyFitnessPal" should be consistent throughout the article, e.g. see line 84.
We have corrected the typo in line 84.
- Paragraph 2.1. can be accompanied by a table with all the descriptive dataset statistics and an example (I mentioned before).
Based on the reviewer’s suggestion we have created Table 1 which presents the demographic statistics of all the users. The Table also presents the demographic statistics of the final subset of users who were considered for analysis.
- I appreciate the notation and the measures. However, it isn't easy to read without an example, e.g., differentiating between dishes and meals.
Based on the reviewer’s suggestion, we use the example in Table 2 in the methodology section to explain the notations (lines 233-239).
- I do not understand why the criterion for the minimal span is 1/2 in equations (1) and (2).
By introducing span as a constraint in equations (1) and (2), we ensure that food-items/ meals that occur globally in the sequence are selected as recurrent. The span is the normalized difference in days between the first and last occurrence of the item/meal in the sequence. The minimal span threshold ½ is chosen so that items that occur in the first half and reoccur in the second half of the sequence are selected. For example, we must consider at least 14 days (2 weeks) to check if items meet the frequency criterion (1/7). With less than a week of data, all items would meet this criterion. Now items that were repeated within the first week but not in the second week would also meet the frequency criterion. With the minimal span of ½, we are selecting items that repeat in the second week.
- The idea of incorporating a span for measuring food recurrence is compelling.
We thank the reviewer for their comment. We hope our explanation of the span and minimal span criterion above showcases the importance of including span in recurrence computation.
- Often citations are not preceded by a space, e.g. see line 346.
We thank the reviewer for highlighting the typos. We have fixed the spacing throughout the paper.
- The text from line 418 is hard to read, and it is better to accompany it with a table. What is this indicator in parenthesis?
We have presented the results discussing the statistics of the recurrence strength measures in a tabular form based on the reviewer’s suggestion in Table 8. The indicator in parenthesis refers to the standard deviation. We have added a line before the presentation of results to clarify the notation.